# Accelerating ice flow at the onset of the Northeast Greenland Ice Stream

**Aslak Grinsted** [1] ✉, **Christine S. Hvidberg** [1], **David A. Lilien** [2], **Nicholas M. Rathmann**[1], **Nanna B. Karlsson** [3], **Tamara Gerber** [1], **Helle Astrid Kjær** [1], **Paul Vallelonga** [4] & **Dorthe Dahl-Jensen**[1,2]

Mass loss near the ice-sheet margin is evident from remote sensing as frontal retreat and increases in ice velocities. Velocities in the ice sheet interior are orders of magnitude smaller, making it challenging to detect velocity change. Here, we analyze a 35-year record of remotely sensed velocities, and a 6-year record of repeated GPS observations, at the East Greenland Ice-core Project (EastGRIP), located in the middle of the Northeast-Greenland Ice Stream (NEGIS). We find that the shear margins of NEGIS are accelerating, indicating a widening of the ice stream. We demonstrate that the widening of the ice stream is unlikely to be a response to recent changes at the outlets of NEGIS. Modelling indicates that the observed spatial fingerprint of acceleration is more consistent with a softening of the shear margin, e.g. due to evolving fabric or temperature, than a response to external forcing at the surface or bed.

Ice-sheet mass loss, a primary driver of sea-level rise, is caused by an imbalance between snowfall, melt, and calving. Between 1992 and 2020 the Greenland Ice Sheet lost the equivalent of 13.5 mm of sea-level rise, of which half can be attributed to dynamic mass loss[1]. The Northeast Greenland Ice Stream (NEGIS) is a prominent ice-flow feature in Greenland. It is the only ice stream that extends into the interior of the ice sheet, initiating less than 150 km from the ice divide. NEGIS drains ~12% of the ice sheet into the Nioghalvfjerdsfjorden (79 N), Zachariæ, and Storstrømmen glaciers[2] (Fig. 1). In contrast to other Greenlandic ice streams such as Sermeq Kujalleq (Jakobshavn Isbræ), neither the position nor the width of NEGIS are topographically controlled[3]. Instead, differences in hydrology or substrate, thermal feedbacks[4], or crystal orientation fabric[5] may control the shear-margin position. The latter two mechanisms would not impose a fixed ice-stream width, implying that the shear margins may respond to perturbations in flow—forced by changes at the margin, climate, or perhaps geothermal heat flux[6]. Importantly, all NEGIS outlets have experienced recent, large dynamical changes. Over the last decades, Zachariæ and Nioghalvfjerdsfjorden glaciers have experienced substantial speed ups, and Zachariæ has undergone considerable frontal retreat[7–10]. Storstrømmen experienced three large surges during the 20th century and is currently in a quiescent phase, building up towards a new surge[11,12]. Ice-flow modeling suggests that the retreat at Nioghalvfjerdsfjorden and Zachariæ will continue for at least a century in response to ocean warming[13]. These dynamical changes at the front will propagate further inland as large-scale ice flow and ice geometry adjust to new boundary conditions. The adjustment will, however, not be instantaneous, and constraining the rate is important for disentangling the different causes of observed changes in ice dynamics with implications for sea-level rise estimates. In this paper, we show how far the adjustment has propagated inland, and how NEGIS velocities are changing in the ice-sheet interior but decoupled from coastal ice-flow dynamics.

## Results

We calculate the acceleration from ice velocity maps spanning the period 1985 to 2018 (Fig. 1). For this we use the ITS_LIVE annual ice-velocity maps[14,15] derived from optical feature tracking of Landsat scenes over Greenland. The along-flow acceleration is calculated by separately fitting linear models to the x and y velocities, and then

[1]Niels Bohr Institute, University of Copenhagen, Copenhagen, Denmark. [2]Centre for Earth Observation Science, University of Manitoba, Winnipeg, MB, Canada. [3]Geological Survey of Denmark and Greenland, Copenhagen, Denmark. [4]Oceans Graduate School, The University of Western Australia, Perth, Australia. ✉e-mail: aslak@nbi.ku.dk

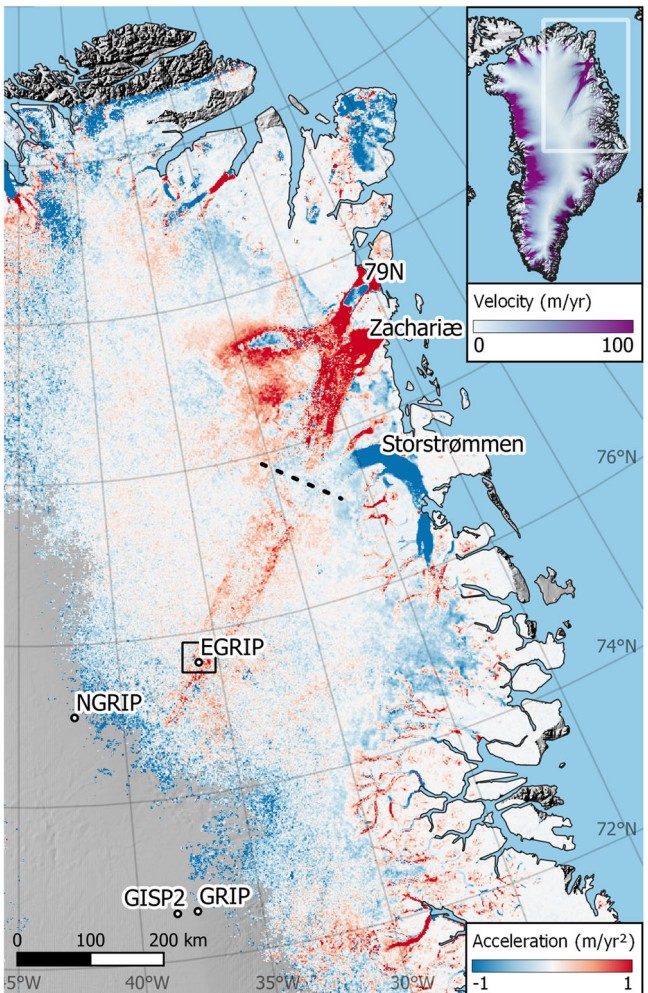

**Fig. 1 | Along flow ice acceleration in North–East Greenland based on 1985–2018 velocity data.** Box near EGRIP indicates location of the stake network in Fig. 2. Grid points with less than 10 years of data has been disregarded. The shear margins of the interior are accelerating, indicating that the ice stream is widening. Dashed line shows region of no clear pattern of acceleration separating inland change from frontal changes. Inset shows overview map colored by ice velocity[29].

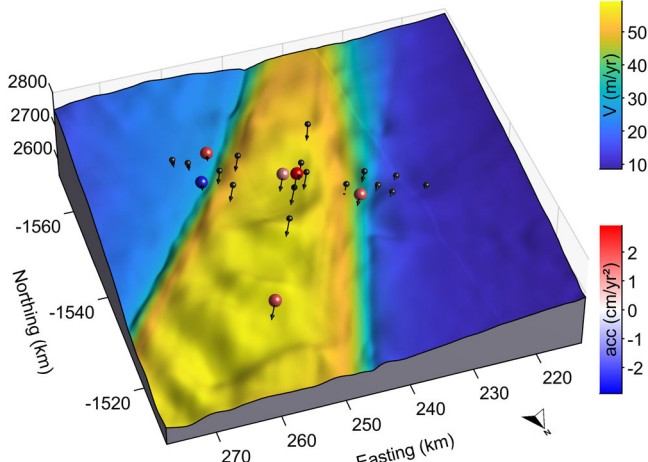

**Fig. 2 | GPS observations of stakes at EastGRIP show accelerating ice flow.** Background map shows ice velocities[29] draped over the landscape. Arrows indicate the velocity and flow direction of each stake. Sphere colors show the estimated acceleration at each stake where it exceeds 1σ (uncertainties shown in online methods Fig. S2).

calculating the along-flow component (see online methods 1). We also use GPS observations from a multi-annual measurement campaign in the vicinity of the EastGRIP deep drilling site (75°38′ N, 36°00′ W, 2700 meters above sea level), located on NEGIS in the interior of the ice sheet (Fig. 1). The measurements include a repeated in-situ survey of a strain net consisting of 63 stakes using the Global Positioning System (GPS). The stakes were observed annually[15] over the period 2015–2019. We estimate the acceleration at each stake with at least three measurements, while correcting for advective acceleration (see online methods 2)[16–18].

We observe large dynamical changes at the three major outlets of NEGIS (Zachariæ, 79 N, and Storstrømmen; Fig. 1). We find, in agreement with ref. 7, that recent large dynamical changes at the major outlets of NEGIS (Zachariæ, Storstrømmen, and 79 N) have already propagated ~100 km upstream. Land terminating glaciers in the region appear to be mostly decelerating (Fig. 1), in accordance with physical theory of thinning glaciers. Importantly, our results also show that the inland region of NEGIS is accelerating along the shear margins, indicating that the ice stream is widening (Fig. 1). Although small, the signal is consistent over several hundreds of kilometers−from the onset of the ice stream to ~77°N. We summarize the spatial acceleration pattern by constructing an average profile across the southeastern shear

margin (see online methods 1) which greatly improves the signal to noise in the remotely sensed data. This shows that the peak acceleration is just on the inside of the shear margin. Stake observations at EastGRIP also show acceleration (Fig. 2) where sufficient data are available. A single exception is a stake placed in an unusually narrow and deep part of the southern shear margin, where coarse spatial resolution limits the quality of the advective contribution. Further discussion of uncertainties can be found in ref. 15 and online methods 2. The stake accelerations are on the order of a few cm/yr² (Fig. 2) and generally lie within the uncertainties of the ITS_LIVE[14] derived estimates. This is unfortunately not very informative as uncertainties on the remotely sensed accelerations are several tens of cm/yr² in the region. Given these uncertainties and the difference in temporal coverage, we caution that the comparison should not be taken as validation, but rather as a lack of disagreement.

Modeling can help us distinguish between different expected responses to hypothesized forcing scenarios. We run four different experiments with an idealized model of a 2D cross section of an ice stream to investigate the surface velocity response to changes in the boundary conditions and ice-flow parameters (see online methods 4)[19–21]. Results are consistent with physical theory, but have greater spatial detail: (1) a thicker ice stream flows faster and but, unlike observations, the velocity anomaly is concentrated in the ice-stream interior; (2) increasing the zone of sliding at the base leads to a spatial map of acceleration that peaks just outside the shear margin; and (3) softening the ice column beneath the shear margin leads to an acceleration that peaks just inside the shear margin.

## Discussion

The observed widening of the ice stream can either be a response to external forcing or an unforced dynamical instability in the streaming flow. Several kinds of external forcing might explain the observations presented here. E.g., it might be supposed that the interior marginal acceleration of NEGIS is a response to the recent large dynamical changes at the outlets. However, the dynamical response of the outlets and the marginal acceleration are separated by a region with no clear patterns of acceleration (Fig. 1, dashed line). This suggests that the interior widening of NEGIS cannot be due to changes at the front, and any links to the recent marginal acceleration is unlikely. This conclusion is supported by kinematic wave theory[22] that leads us to conclude that the signal would be greatly dampened at these distances and

would likely be undetectable at EastGRIP (see online methods 3). The ice-stream widening might, instead (or also), be a response to an ongoing slow adjustment of the ice-sheet geometry to changes further in the past, such as the last glacial termination, as geometry controls the large scale flow of ice and hydrological drainage[23]. Finally, the widening could be a response to recent local changes in climate, such as a trend in accumulation. However, this seems unlikely as accumulation trends in this region are small[24], and likely have been small for the past several hundred years[25]. Our modeling finds that the spatial pattern of acceleration is most consistent with a softening of the ice column beneath the shear margin, rather than changes at the surface or the base. The softening could be facilitated via warming or an evolving fabric. This is unlike the basal mechanisms that have been proposed for the reorganization of ice streams in the Siple coast region of West Antarctica[16]. Given the additional evidence that the interior part of NEGIS is not topographically steered[26], we conjecture that the observed widening of NEGIS is indicative of a dynamical instability in the ice stream.

Further study is needed to understand the mechanism driving the ice-stream widening. A reconstruction of NEGIS frontal fluctuations over the past 45,000 years suggests that relatively large retreats do not have to be associated with large changes in air or ocean temperatures[2]. This could indicate that NEGIS is very susceptible to even slight changes in forcing[2]. However, the observations could also be partly explained by unforced dynamical instabilities in the ice stream. Regardless of the cause, our study shows that the ice sheet interior is more dynamically variable than hitherto measured, which has deep implications for the future of the ice sheet. This study therefore raises pressing questions: Why is the ice stream accelerating so deep in the interior? Could NEGIS widening be a pre-cursor to a large-scale reorganization of ice flow in North East Greenland similar to changes in the ice streams at Siple Coast, Antarctica[27,28]? Our findings highlight the need to understand and carefully separate internal dynamics of ice streams from different forced perturbations and long-term changes in climate forcings. This is crucial if we are to accurately project ice-sheet mass loss in response to recent and future climate change.

## Data availability
All source data are available in refs. 14, 15. The derived acceleration can be found at https://doi.org/10.5281/zenodo.6806677.

## Code availability
Model code can be found at https://doi.org/10.5281/zenodo.6806513.

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

## Acknowledgements

This work was supported by the Villum Investigator Project IceFlow (16572), a Dancea grant from the Danish Environmental Protection Agency (EPA), and Villum Foundation Experiment grant (2361). EastGRIP is directed and organized by the Physics of Ice, Climate and Earth group, at the Niels Bohr Institute, University of Copenhagen. It is supported by funding agencies and institutions in Denmark (A. P. Møller Foundation, University of Copenhagen), USA (US National Science Foundation, Office of Polar Programs), Germany (Alfred Wegener Institute, Helmholtz Centre for Polar and Marine Research), Japan (National Institute of Polar Research and Arctic Challenge for Sustainability), Norway (University of Bergen and Trond Mohn Foundation), Switzerland (Swiss National Science Foundation), France (French Polar Institute Paul-Emile Victor, Institute for Geosciences and Environmental research), Canada (University of Manitoba) and China (Chinese Academy of Sciences and Beijing Normal University).

## Author contributions

A.G. analyzed data. A.G., C.S.H., D.A.L. and N.M.R. conceived study. A.G., N.M.R., D.A.L. and T.G. modeled ice flow. A.G., C.S.H., N.B.K., H.A.K., P.V. and D.D.J. collected data. All authors interpreted results and wrote paper.

## Competing interests

The authors declare no competing interests.
