## [Peer Review File · Nature Communications]

Accelerating ice flow at the onset of the Northeast Greenland ice streamReviewers' Comments:

Reviewer #1:

Remarks to the Author:

This is a tightly organized and focussed short paper that reports new observations suggesting that the far-upstream portions of the North East Greenland Ice Stream (NEGIS) is changing its flow pattern (widening) and thus accommodating accelerating ice flow. The data are compelling and the analysis is both on target and satisfactory.

The manuscript is well written, well organized and well proof-read; hence, there is very little to suggest in the way of improvements or things to check or explain better. Some comments that I provide:

Figure 1 - I found this figure, because of the double color bar, a bit hard to understand. I wonder if it would be better if there were two panels, one for each color bar... Plus, in the caption, or in the body, it might be worth taking a few sentences to explain an example of what a reader should see in that figure (this is done to some extent, however, it may be worth repeating in the caption itself).

line 38 - It may not be clear to all readers that Zachariah and Nioghavfjorden are outlets for the NEGIS (I didn't know this until I was reading down at line 58). It may be best to put the sentence on line 58 around line 38 plus make a reference to figure 1 around line 38.

line 93- I would prefer the word "more variable" to "more dynamical", but both are OK.

line 96- It may be worth adding that it is important to understand also the "internal dynamics" of the ice stream (which I think was what was meant by "dynamic response").

general - Could a reference be made at some point to the slow-down of the Kamb Ice Stream in Antarctica, and/or to the possibility that Heinrich events are a result of internal ice-stream instability/dynamics?

Reviewer #2:

Remarks to the Author:

The manuscript "Accelerating ice flow at the onset of the Northeast Greenland ice stream" by A. Grinsked et al. presents observations of ice flow acceleration in the interior of the Greenland Ice Sheet, close to the onset of the North East Greenland Ice Streams. This acceleration highlights a widening of this ice stream, unrelated to changes already observed close to the ice sheet fronts, and the authors suggest that this acceleration might reflect an instability of the ice stream.

Given the relatively low velocities in the interior of the ice sheet that are within the error bounds of most observations, estimating changes and a possible acceleration in this area of the ice sheet is a challenging task. By showing consistent results obtained with remote-sensing estimates as well as GPS observations, the authors provide convincing evidence that signal is real. However, this acceleration is very small (~ 2 cm/yr when the ice stream velocity reaches >40 m/yr in this region, so $\sim 0.5\%$) and there is little to no discussion on the uncertainty associated with these two sources of observations. The paper includes a reference to Hvidberg et al., 2020, but given the potential importance of uncertainty in this context, I would have like to see an in-depth discussion of this subject in the main text of the manuscript.

Furthermore, while several mechanisms that might cause the acceleration, such as propagation of changes happening at the front of the glacier, are ruled out by the authors, there are very few mechanisms proposed to explain a cause for this inland acceleration. Only a general conjecture

suggesting a dynamical instability of the ice stream is proposed, without further detail on the cause of this instability. I would like to see this hypothesis explored further, mechanisms proposed and tested with analytical solution, simple models, ... to validate the order of magnitude of this acceleration and this manuscript do more than simply ask questions.

In conclusion, I think this manuscript presents a nice new observation of changes far inland in the Greenland Ice Sheet, but it lacks a better discussion of possible explanations for this acceleration, backed up by simple analysis or modeling to demonstrate that the mechanisms proposed to explain this acceleration are consistent with the observations shown.

Reviewer #1 (Remarks to the Author):

This is a tightly organized and focussed short paper that reports new observations suggesting that the far-upstream portions of the North East Greenland Ice Stream (NEGIS) is changing its flow pattern (widening) and thus accommodating accelerating ice flow. The data are compelling and the analysis is both on target and satisfactory.

The manuscript is well written, well organized and well proof-read; hence, there is very little to suggest in the way of improvements or things to check or explain better. Some comments that I provide:

Figure 1 - I found this figure, because of the double color bar, a bit hard to understand. I wonder if it would be better if there were two panels, one for each color bar... Plus, in the caption, or in the body, it might be worth taking a few sentences to explain an example of what a reader should see in that figure (this is done to some extent, however, it may be worth repeating in the caption itself).

We have moved the velocity legend immediately next to the velocity overview inset map, and expanded the caption slightly.

line 38 - It may not be clear to all readers that Zachariah and Nioghavfjorden are outlets for the NEGIS (I didn't know this until I was reading down at line 58). It may be best to put the sentence on line 58 around line 38 plus make a reference to figure 1 around line 38.

We now state that these three glaciers drain NEGIS in line 31. We have also added a reference to the map figure at this point (fig 1).

line 93- I would prefer the word "more variable" to "more dynamical", but both are OK.

DONE.

line 96- It may be worth adding that it is important to understand also the "internal dynamics" of the ice stream (which I think was what was meant by "dynamic response").

Correct. That was our intent. This sentence was not 100% clear. We have rephrased and now explicitly state "internal dynamics".

general - Could a reference be made at some point to the slow-down of the Kamb Ice Stream in Antarctica, and/or to the possibility that Heinrich events are a result of internal ice-stream instability/dynamics?

Good suggestion. We now refer to the Kamb/Siple coast region in two places:

- 1) we point out that the changes we observe are most consistent with a softening of the ice column. This is unlike the basal mechanism proposed for Kamb and neighbors .
- 2) We've added an open question whether the NEGIS widening is a precursor to a large scale ice flow reorganization as seen in west antarctica.

Reviewer #2 (Remarks to the Author):

The manuscript "Accelerating ice flow at the onset of the Northeast Greenland ice stream" by A. Grinsted et al. presents observations of ice flow acceleration in the interior of the Greenland Ice Sheet, close to the onset of the North East Greenland Ice Streams. This acceleration highlights a widening of this ice stream, unrelated to changes already observed close to the ice sheet fronts, and the authors suggest that this acceleration might reflect an instability of the ice stream.

Given the relatively low velocities in the interior of the ice sheet that are within the error bounds of most observations, estimating changes and a possible acceleration in this area of the ice sheet is a challenging task. By showing consistent results obtained with remote-sensing estimates as well as GPS observations, the authors provide convincing evidence that signal is real. However, this acceleration is very small (~2 cm/yr when the ice stream velocity reaches >40 m/yr in this region, so ~0.5%) and there is little to no discussion on the uncertainty associated with these two sources of observations. The paper includes a reference to Hvidberg et al., 2020, but given the potential importance of uncertainty in this context, I would have like to see an in-depth discussion of this subject in the main text of the manuscript.

We have added a new figure to 'online methods 1' that helps address the statistics of the remote sensing acceleration. In this figure we construct an average profile of the acceleration signal across the shear margin. This large-scale spatial averaging improves the signal-to-noise ratio and shows that the acceleration signal far exceeds the standard error.

While we would have liked to add this new figure to the main text, and perhaps also include a more extensive discussion of the statistics/uncertainties there, the journal format precludes this option. We have put this in the supplementary online material in line with journal style and instructions.

Furthermore, while several mechanisms that might cause the acceleration, such as propagation of changes happening at the front of the glacier, are ruled out by the authors, there are very few mechanisms proposed to explain a cause for this inland acceleration. Only a general conjecture suggesting a dynamical instability of the ice stream is proposed, without further detail on the cause of this instability. I would like to see this hypothesis explored further, mechanisms proposed and tested with analytical solution, simple models, ... to validate the order of magnitude of this acceleration and this manuscript do more than simply ask questions.

Thank you - This is a good suggestion. We therefore now include a model of an idealized ice stream (online methods 4) to investigate which proposed mechanism is most plausible (or most consistent with observations). We feel this has been a very useful addition to the study. It clearly adds more weight to some explanations rather than others. We have tried to keep the observational focus of the study, so our idealized cross profile modelling is an appropriate level of modelling detail.

In conclusion, I think this manuscript presents a nice new observation of changes far inland in the Greenland Ice Sheet, but it lacks a better discussion of possible explanations for this acceleration, backed up by simple analysis or modeling to demonstrate that the mechanisms proposed to explain this acceleration are consistent with the observations shown.

We hope that the modelling added in our revision is satisfactory support for our interpretation.

Reviewers' Comments:

Reviewer #2:

Remarks to the Author:

The revision of the manuscript "Accelerating ice flow at the onset of the Northeast Greenland ice stream", by A. Grinsked et al., includes new discussion on the uncertainty of the observations, both in the main text and the supplement, as well as some idealized mode results to investigate possible sources of changes, in response to the reviewer's comments. These new additions allow to better understand potential reasons of this acceleration, add depth to the manuscript and provide more evidence of the potential impact of the observations provided. While more realistic simulations would allow to better represent important processes (enhanced deformation with shear, ice temperature, etc.), this is beyond the point of the current manuscript that mostly aim to show this intriguing pattern of acceleration inland the Greenland ice sheet, far from the regions usually affected by changes. I therefore only have minor comments (listed below). I think it would also be important to add the reason suspected for these changes to appear in the abstract, and to tone it down a little in the main text (l.105) given the limitations of the experiments done, as this should be further confirmed with other experiments and additional observations.

Minor comments (line numbers refer to the version with track changes):

l.73: Add reference for ITS_LIVE

l.78: Add that the model is a 2d cross-section of the ice stream

l.81: "faster and but that the velocity" -> rephrase

l.84: What are the implications?

l.87: "must" -> "can"

l.105: I see this dynamic instability to be one possible consequence so I would mention this is only one option

Fig.1 caption: "accelerating, indicating that"

Replies to reviewers comments

Reviewers comments appear in indented italics. Our responses appear below each comment.

Reviewer #2 (Remarks to the Author):

The revision of the manuscript “Accelerating ice flow at the onset of the Northeast Greenland ice stream”, by A. Grinsted et al., includes new discussion on the uncertainty of the observations, both in the main text and the supplement, as well as some idealized mode results to investigate possible sources of changes, in response to the reviewer’s comments. These new additions allow to better understand potential reasons of this acceleration, add depth to the manuscript and provide more evidence of the potential impact of the observations provided. While more realistic simulations would allow to better represent important processes (enhanced deformation with shear, ice temperature, etc.), this is beyond the point of the current manuscript that mostly aim to show this intriguing pattern of acceleration inland the Greenland ice sheet, far from the regions usually affected by changes. I therefore only have minor comments (listed below). I think it would also be important to add the reason suspected for these changes to appear in the abstract, and to tone it down a little in the main text (l.105) given the limitations of the experiments done, as this should be further confirmed with other experiments and additional observations.

Thank you.

We have added a sentence to the abstract that highlights the conclusions we can draw from our model experiments. With this additional sentence then we have reached the abstract word limit.

Minor comments (line numbers refer to the version with track changes):

l.73: Add reference for ITS_LIVE

Done.

l.78: Add that the model is a 2d cross-section of the ice stream

Done.

l.81: “faster and but that the velocity” -> rephrase

Fixed.

l.84: What are the implications?

Line 84 is intended as a relatively pure results section. I.e. we leave the discussion and implications for later. Specifically, we talk about the implications later in the final sections of the manuscript. We try not to overstate the implications, but here in the response we can be a bit more free in our interpretation. Let us suppose, for the sake of argument, that the observed signal really is due to a

dynamical instability involving a feedback between flow and an evolving fabric. In that case it seems clear that it will be near-impossible to model the ice stream margin position without a model that accounts for the fabric evolution. Further, if the present day ice-stream position is a consequence of the accumulated dynamical margin evolution, then this might explain why it is exceedingly hard to get ice sheet models to have a good representation of NEGIS without tuning the sliding with that goal.

I.87: "must" -> "can"

Done

I.105: *I see this dynamic instability to be one possible consequence so I would mention this is only one option*

This concerns the sentence: "*Given the additional evidence that the interior part of NEGIS is not topographically steered²⁰, we conjecture that the observed widening of NEGIS is indicative of a dynamical instability in the ice stream.*"

We feel that "conjecture" and "indicative" are sufficiently equivocal that this phrasing is justified at this point. We do not ignore other possibilities, which are discussed immediately prior to this sentence. We have therefore not made any revisions here.

Fig.1 caption: "*accelerating, indicating that*"

Done. Thank you.